# Robust Detection of Directional Adversarial Attacks in Deep Neural Networks for Radiological Imaging

## Abstract

Deep learning is now central to radiology, helping detect changes on X-rays, CTs, and MRIs. However, these systems are highly vulnerable to adversarial attacks - small, crafted perturbations that mislead models while appearing unchanged to humans. Such errors risk missed cancers or false positives. Studies show over 90% attack success on medical scans. Almost 30% of US hospitals use AI in imaging in 2023. Therefore, an effective method of detecting such attacks and their directionality is crucial to ensure the credibility and reliability of medical systems based on DNNs. We propose a novel detection framework that leverages subsequent attacks using random noise to identify adversarial perturbations. Our approach uses the analysis of variations in the Sigmoid function to distinguish genuine medical images from adversarially manipulated ones. Specifically, we compare prediction differences between clean and attacked images, as well as between different adversarially altered versions, to improve detection accuracy. We evaluated our method on three popular medical datasets including 'Chest X-Ray Images for Classification', 'Retinal Fundus Multi-disease Image Dataset' and 'Brain Tumor Dataset' under various attack scenarios, including random noise. Our framework achieved up to 99.8% ACC in identifying adversarial directional attacks, significantly outperforming existing defense mechanisms in terms of detection accuracy. It consistently identified adversarial samples across varying attack strengths while keeping false positives low, showing strong reliability and potential for clinical use. Furthermore, our proposal highlights its potential for real-world deployment.

## 1 Introduction

In recent years, the digitization of healthcare and the widespread adoption of Picture Archiving and Communication Systems (PACS) have transformed the way medical images are stored, accessed, and transmitted. Medical image analysis has seen remarkable progress, especially in the areas of segmentation and classification. AI-driven models are increasingly used in healthcare, among others, helping radiologists detect pneumonia on chest X-rays Mujahid et al. (2024), diagnosing brain tumors from MRI scans Lamrani et al. (2023), and allowing early detection of retinal diseases Parmar et al. (2024). According to IDTechEx, the market for AI medical image diagnostics is expected to surpass $3 billion by 2030 in five segments, including cancer and cardiovascular disease, respiratory, retinal, and neurodegenerative diseases IDTechEx (2020).

However, this shift has also introduced new cybersecurity vulnerabilities. Radiological images, which are critical for diagnosis and treatment planning, are increasingly being targeted by cyberattacks ranging from data breaches to image manipulation and adversarial attacks on AI-based diagnostic tools. These threats not only endanger patient privacy, but also pose serious risks to clinical decision-making and patient safety.

### 1.1 Background and motivation

As hospitals and clinics increasingly rely on digital systems to store and transmit medical images, radiology has become a prime target for cyberattacks. More than 90% of healthcare institutions now use Picture Archiving and Communication Systems (PACS), yet many of these systems lack

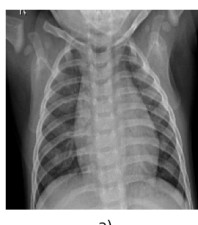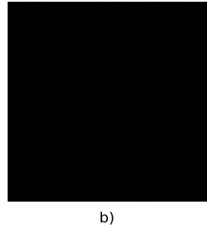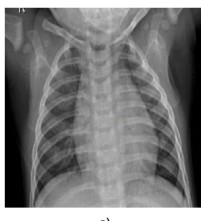

a)                                b)                                c)

Figure 1: Visualization of image perturbations: a) original Chest X-ray image, b) adversarial noise, c) The original image combined with the perturbation, demonstrating no visible differences.

modern security features. In 2022 alone, more than 28 million patient records were exposed in healthcare-related data breaches in the US, with imaging data often among compromised content. Research has shown that DNNs are particularly vulnerable to adversarial manipulations, and even small perturbations can manipulate the predictions of the model while leaving the image visually unchanged Pauling et al. (2022). As shown in Fig. 1, the noise is almost invisible to the human eye but can still change the prediction and diagnosis performed by the DNN model. A misdiagnosis due to an adversarially perturbed image can result in incorrect treatment decisions.

## 1.2 ADVERSARIAL ATTACKS IN MEDICAL IMAGING

Adversarial attacks take advantage of the non-linearity and high-dimensional decision space of DNNs by introducing small but structured perturbations that may change the model predictions. These attacks can be categorized into white-box and black-box attacks. In white-box attacks, the adversary has full access to the model architecture and parameters, while in black-box attacks, the model transferability is based on adversarial examples generated for one model that can influence another without requiring direct access to its parameters Zhang et al. (2020).

The adversarial attack techniques include among others (1) Fast Gradient Sign Method (FGSM) Wong et al. (2020) which is a single-step attack that perturbs input images along the gradient direction to maximize the classification error; (2) Projected Gradient Descent (PGD) Gupta et al. (2017) which is an iterative version of FGSM that refines perturbations over multiple steps for stronger adversarial effectiveness; and (3) Carlini-Wagner attack (CW) Sarkar et al. (2024) which is a sophisticated optimization-based attack that minimizes perturbations while ensuring misclassification, making detection particularly difficult.

Although adversarial attacks pose a risk to medical imaging AI systems, one of the most critical is the directional adversarial attack Yu et al. (2024). Unlike traditional adversarial attacks, which aim to degrade overall model performance, directional attacks manipulate predictions toward a specific incorrect class. For example, an attacker could manipulate a scan of a malignant tumor to make the model classify it as benign, potentially resulting in missed cancer diagnoses. Similarly, a chest X-ray image or fundus photography could be adversarially modified to appear normal, increasing the risk of undiagnosed disease.

Recent studies have shown that adversarial directional attacks are not only highly effective but also difficult to detect Tsai et al. (2023). For example, an attack on dermoscopic images as well as lung CTs successfully misclassified adversarially changed images with a success rate of 99% while not detected by radiologists. Similarly, adversarial attacks led to the misclassification of the brain tumor and the accuracy dropped from 87% to below 12% Mirsky et al. (2019). Such attacks raise serious concerns about the safety of AI-driven diagnostics in real-world clinical applications.

## 1.3 CHALLENGES IN DETECTING ADVERSARIAL ATTACKS IN MEDICAL IMAGING

Despite increasing awareness of adversarial threats, existing defense mechanisms have proven to be insufficient in medical imaging applications. Standard adversarial detection methods, such as adversarial training and input preprocessing, have shown limited success due to the high dimensionality and domain-specific features of medical images Dong et al. (2023).

Adversarial training in medical image analysis is crucial because medical images differ significantly from regular images, requiring specialized attack and defense strategies. The main challenges include: (1) Modality-specific factors appear as different imaging techniques (MRI, CT, X-ray, ophthalmoscopy, ultrasound) have unique characteristics such as intensity distributions and channel configurations; (2) Limited and imbalanced datasets make models more vulnerable to adversarial manipulation; (3) Clinical impact and ethical concerns increase the stakes, as adversarial attacks in medical AI could lead to severe misdiagnoses, directly affecting patient care; (4) Anatomical structures and specialized medical knowledge make defense even more challenging, as attacks can subtly alter key features, increasing the risk of misdiagnosis. These challenges emphasize the need for robust and clinically aware defenses against adversarial attacks Dong et al. (2023).

Given these challenges, in this study, we propose a novel detection framework specifically designed to identify directional adversarial attacks in medical imaging AI systems that take advantage of the subsequent attack using random noise to identify adversarial perturbations. Our approach integrates the prediction differences between clean and noisy clean images and attacked and noisy attacked images to differentiate genuine medical images from adversarially manipulated ones.

In conclusion, our main contribution is to conduct a comprehensive analysis of directional adversarial attack strategies in medical imaging, highlighting their impact on DNN diagnostic models. We develop a robust adversarial detection method by analyzing prediction differences across multiple scenarios: (1) clean images versus those perturbed with random noise and (2) clean images versus those that have been both adversarially attacked and then re-attacked after adding random noise. By comparing these variations, we aim to distinguish adversarial manipulations from natural perturbations, improving detection accuracy. At the same time, we conduct an extensive evaluation on the medical images benchmark datasets such as X-ray, MRI and fundus photography for disease detection with deep learning under various attack scenarios, demonstrating the effectiveness of our proposed detection framework. We also provide quantitative evidence of the impact of adversarial attacks on network performance by analyzing prediction differences.

### 1.4 RELATED WORKS

Adversarial attacks have emerged as a critical threat to DNNs as described by Szegedy in his famous work Szegedy et al. (2013). In Bhambri et al. (2019) showed that small, imperceptible perturbations can lead to significant misclassification. Goodfellow et al. (2014) introduced the Fast Gradient Sign Method, leading to increased research on adversarial attacks and defense strategies in deep learning.

Medical AI models have been shown to be particularly vulnerable to adversarial attacks. Finlayson et al. (2019) demonstrated that perturbations could lead to misclassification in skin cancer and chest X-ray diagnosis models with success rates exceeding 90%. Ma et al. (2019) further explored adversarial robustness in MRI-based classification, showing that attacks could reduce model precision from 95% to below 10%. Adversarial attacks on ophthalmology models have resulted in misclassifications in the detection of retinal diseases Abbas et al. (2024), highlighting the need for robust security measures in clinical AI systems. Various defense strategies have been proposed to mitigate adversarial threats in deep learning. Adversarial training, where models are trained on both clean and adversarially perturbed samples, is one of the most widely adopted defenses Madry et al. (2017). However, this approach is computationally expensive and often leads to a performance drop in clean samples. A decent level of detection has been achieved by using a second model for the same task Guo et al. (2019), the results are high only for attacks with high epsilon, which is mostly visible to the human eye and requires training of a second model that can also be attacked. Recent research has explored statistical feature analysis Ma et al. (2018), Bayesian uncertainty estimation Begoli et al. (2019), and adversarial detection networks Lee et al. (2018) as potential solutions. However, these techniques have not been extensively evaluated in medical imaging contexts, where unique challenges such as data heterogeneity and regulatory constraints must be considered.

## 2 METHODOLOGY

The proposed framework shown in Fig. 2 evaluates the impact of adversarial attacks on image classifications by introducing random noise, generating adversarially perturbed images, and analyzing prediction differences using deep learning models to assess classification robustness and inter-

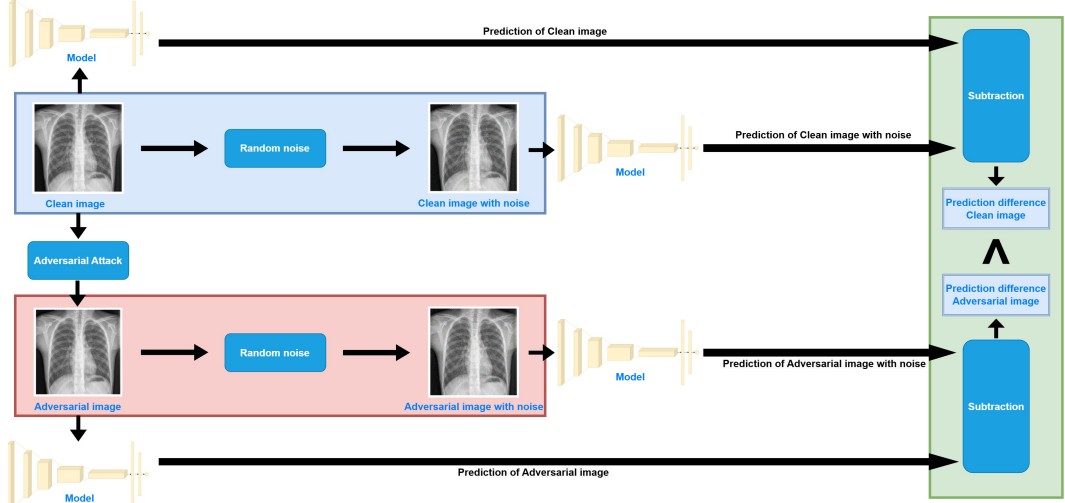

Figure 2: Overview of the adversarial detection framework. (1) Clean images are processed, and random noise is added to generate noisy images. (2) Adversarial attacks are applied to the clean and random noise is added to generate noised adversarially dataset. (3) A deep learning model trained and evaluated on clean dataset, make predictions for all created datasets, and then prediction differences and interpretability metrics are analyzed to assess detection of adversarial attacks.

pretability. By attacking an already perturbed image, we can change the class detected by the model. However, it is crucial that the attack redirects the prediction to the correct class rather than an arbitrarily distributed one. To achieve this, the attack must target the perturbation itself rather than the image, ensuring that the class does not shift unpredictably. Moreover, in the case of a correctly classified clean image, the attack should not alter a valid prediction. This implies that the attack must be subtle, similar to directed attacks like FGSM or PGD, where perturbations are typically within 1%.

## 2.1 Attack Formulation and Mathematical Model

The attack used is to perturb only the changes introduced by the perturbations resulting from the adversarial attack while keeping the image itself intact. The ideal neutralizing attack turns out to be a simple random noise . It does not change the model's predictions for any specific class so it is neutral; it will simply cause some change, and due to its ineffectiveness in attacking adversarially at a low percentage of perturbations, it will not cause a significant change in the clean image.

However, it is important to consider when random noise becomes effective. When the noise level is sufficiently high, around 16% or more, making it visible to the human eye, it begins to significantly influence the model prediction. Generally, stronger perturbations lead to greater shifts in classification. The most effective scenario would be if the noise completely covered the image; however, in this case the goal is to target the perturbation itself. Introducing a small amount of random noise (a few percent) does not substantially alter the prediction of a clean image, but can effectively neutralize the impact of an adversarial attack. This ensures that the original classification remains largely unchanged while mitigating adversarial modifications.

## 2.2 Detection framework

In most attacks, the image is modified so that the model prediction is directed to a specific target. In this case, the image is attacked with random noise at a low intensity (a few percent). This level of noise represents a significant change for the perturbation itself, but only a minor alteration for the overall image. After applying the noise, the prediction of the model is evaluated. If the image was not previously subjected to an adversarial attack, the prediction will shift only slightly, similar to a weak random noise attack. It is also important to note that this change can occur in any direction, as the noise is not designed to push the prediction toward a specific outcome.

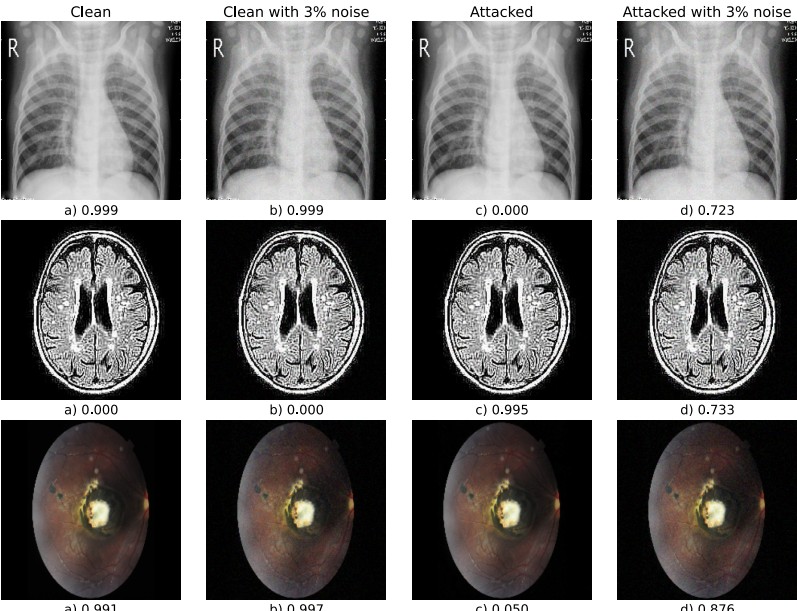

Figure 3: Impact of noise and adversarial attacks on Sigmoid performance: a) the original image, b) the image with 3% random noise, c) the adversarially attacked original image, and d) the adversarially attacked image with 3% random noise. Below the images, the Sigmoid function outputs illustrate how noise and adversarial attacks affect the model's confidence in its predictions.

However, if the image was previously attacked, the prediction will change. The effect of the perturbation resulting from the directed adversarial attack will be significantly reduced, resulting in a prediction that is decidedly closer to the original image. It is important to note that the attacked image almost always predicts the same class as the original clean image. This suggests that after adding random noise, the prediction shifts back toward the original class, providing strong evidence that the image was not previously adversarially manipulated. Fig. 2 illustrates the methodology, which involves comparing the model's predictions before and after adding random noise to assess the differences in classification.

## 2.3 FEATURE EXTRACTION FOR ADVERSARIAL DETECTION

It should be noted that if adding noise to an image causes the prediction to shift back toward its originally detected class or only slightly toward another class, it can be concluded that the image was not previously subjected to an adversarial attack. On the other hand, if the prediction has changed significantly toward another class, then we are dealing with an attacked image, and the actual class will be the one toward which the prediction has changed. The last step to detect adversarial attacks is to set an appropriate threshold of the prediction difference that will separate as many attacked images from as many clean images as possible.

## 2.4 IMPLEMENTATION DETAILS AND HYPERPARAMETER SELECTION

To achieve detection effectiveness, the attack strength must be appropriately adjusted to the given problem. The attack must be weak enough not to significantly change the prediction of clean images and, at the same time, strong enough to eliminate the changes introduced by the directed adversarial attack. Noise added with the appropriate strength will allow for precise separation of the attacked images from the unattacked ones in the distribution. Then, according to the previously obtained distribution, the threshold of the prediction difference should be selected. The best effect will be around the marginal values of the prediction difference between clean cases, which will allow for proper separation of clean images from the attacked ones. So, the best option for each model will be to manually select the threshold, while a good approximation for each model will be a well-known

statistical threshold $\theta = 2 \cdot \text{std}(t)$, often used to capture most of the variation in the normally distributed data, where $t$ denotes the differences between the predictions of the model in clean images and their noisy counterparts. The standard deviation quantifies the spread of these differences. Multiplying it by 2 corresponds to a common statistical rule, where approximately 95% of values drawn from a normal distribution lie within two standard deviations of the mean. Therefore, the threshold $\theta$ is intended to capture significant deviations beyond typical noise-induced variation.

## 2.5 MODELS

The pretrained ResNet-50V2, InceptionV3 and ViT-B16 models, originally trained on the ImageNet dataset, were used as feature extractors for our two-class classification task. The final classification layers (model heads) were removed and replaced with a sigmoid activation layer to match the binary nature of our dataset. Initially, all layers of the pretrained models were frozen to preserve the weights learned during ImageNet training. Subsequently, a few of the final layers were unfrozen to allow fine-tuning, enabling the model to adapt its high-level representations to the specific characteristics of our dataset. The models were effectively trained, achieving good ACC and recall average scores presented in Table 1. Such capabilities are also confirmed by the high ROC-AUC value.

Table 1: Accuracy | Recall | AUC of the prepared models

| Pretrained model | Chest X-Ray | Tumor MRI | Retinal fundus |
|---|---|---|---|
| ResNet50V2 | 0.94 \| 0.94 \| 0.99 | 0.99 \| 0.99 \| 1.00 | 0.91 \| 0.91 \| 0.95 |
| InceptionV3 | 0.92 \| 0.92 \| 0.99 | 0.99 \| 0.99 \| 1.00 | 0.90 \| 0.90 \| 0.95 |
| ViT-B16 | 0.91 \| 0.91 \| 0.97 | 0.99 \| 0.99 \| 1.00 | 0.89 \| 0.89 \| 0.96 |

## 3 RESULTS AND ANALYSIS

To benchmark our results, we performed evaluations on three publicly available medical datasets.

### 3.1 DATASETS

#### 3.1.1 CHEST X-RAY IMAGES FOR CLASSIFICATION

'Chest X-ray Images for Deep Learning Pneumonia Detection' is a widely used dataset to train and evaluate deep learning models in medical image analysis Kermany et al. (2018). It consists of labeled chest X-ray images categorized into pneumonia and normal (healthy) cases. The collection shows appropriately selected X-ray images of children's lungs, divided into images of healthy lungs and those with inflammation. This dataset contains 5856 reviewed chest X-ray images. The images are divided into a training set of 3883 pneumonia and 1349 healthy examples and a test set of 390 pneumonia and 234 normal cases, respectively.

#### 3.1.2 RETINAL FUNDUS MULTI-DISEASE IMAGE DATASET

The Retinal Fundus Multidisease Image Dataset (RFMiD) covers a wide range of diseases that occur in routine clinical settings Pachade et al. (2020). The RFMiD consists of 3200 fundus images taken with three different fundus cameras with 46 conditions described by consensus of two senior retinal experts. The images are divided into a training set containing 1920 images, a test set consisting of 134 healthy images and 506 examples with any disease, and a validation set containing 640 images.

#### 3.1.3 BRAIN TUMOR DATASET

The Brain Tumor MRI dataset provides a comprehensive collection of labeled brain MRI scans Mandal (2024) that mainly includes a dataset widely used as a benchmark for the development and evaluation of deep learning models Cheng (2017). It comprises magnetic resonance imaging (MRI) scans labeled in three types of tumor: meningioma, glioma, and pituitary tumor. The dataset classes have been converted to healthy and tumor. Brain Tumor MRI dataset includes contrast-enhanced images collected under consistent clinical conditions with a total of 7023 images.

## 3.2 EXPERIMENTS

Each of the test sets of the model was attacked using four attack methods. Fast Gradient Sign Method (FGSM), Basic Iterative Method (BIM), Projected Gradient Descent (PGD) with an iteration change strength of 0.001 to 0.01, and DeepFool (DF) with a maximum of fifty iterations and prediction overshoot of 0.02. The attack stopped changing strength or stopped iterating when the attacked image changed the class predicted by the model to the opposite one. It is worth mentioning how effective adversarial attacks can be, PGD and BIM were able to change the predicted class of 93-100% of the images. FGSM was able to precipitate a prediction of 71% of images, and the precision of the change in DF prediction was 99% on ResNet50V2-based and ViT-B16 models, but was much worse with InceptionV3-based models for which the precision was 62.6%.

The experiment only used images from test sets whose model did not mispredict because an attack would not require changes in the image. Then an attack was launched on each of the images so that they were misclassified by the model. The method presented here is based on detecting the difference in the prediction in the attacked images, and given that not every attack was successful, that is, the image prediction did not change noticeably after the attack, the attacked images had to be discarded. Then, random noise was added to rest of the images and the difference in prediction between images before and after adding noise was checked. The noise power was checked from a value of 1% to a value of 5% with a step of 1%. Each of the noise additions was repeated eight times and took the median for greater confidence in the results and to avoid extreme outliers in the event of similarity to the gradient of any of the classes.

The most effective universal power for all types of models was 3%. This value best meets the previously mentioned assumptions, i.e. effectively eliminates changes resulting from the adversarial attack, and at the same time does not significantly change the prediction of clean images but each model is different, just as the adversarial attacks and their power on each model were slightly different. It is noteworthy that for the FGSM, BIM and PGD attacks the strength of the random noise was different for the best results than for DeepFool. This is probably due to the fact that DeepFool tries to find the smallest possible perturbation that will result in a class change. So, the added changes to the image on average were smaller than for the other types of attack, resulting in a best epsilon of random noise for each model attacked by DeepFool was 0.01, because it was enough to eliminate changes from adversarial attack while at the same time changing the clean image prediction the least. Looking at the remaining adversarial attacks such as FGSM, BIM, PGD, the intensity of which was limited in the range 0.001 - 0.01 epsilon, and for each model for these three adversarial attacks, the best results came out for random noise of the same power. The best results for the all models trained on the Chest X-ray dataset were achieved with a random noise epsilon of 0.05. For the Brain MRI dataset, the best result was obtained with an epsilon of 0.05 for ResNet50V2 and 0.03/0.04 for InceptionV3 and ViT-B16. For the eye fundoscopy dataset, the best result for ResNet50V2 was with an epsilon of 0.03 for InceptionV3 and ViT-B16. The difference between the models and, at the same time, the similarity in the epsilon of the random noise for the different adversarial attacks is due to the different resilience of each model and the type of images that directly determine the strength of adversarial attacks to be effective.

This method also performs well compared to the well-known method such as the Mahalanobis distance Lee et al. (2018). Mahalanobis was tested using three layers of models based on ResNet50V2 and InceptionV3, while in the case of ViT-B16, only the dense layer was used due to excessive computational complexity. Using the Mahalanobis score, 92.32% of adversarial attacks were detected, with false positives detected at a level of 5%. Mahalanobis performed well in detecting typical white-box attacks, but it remains vulnerable to adaptive attacks that can adapt to the extracted features on which detection is based. Considering such a case in a method based on adding random noise to reduce the impact on the model, which causes differences in prediction, the person carrying out the attack has only two options. Either not to make the prediction change (where this is the attacker's main goal), or to make the addition of random noise not affect the change caused by the attack, or at least not significantly. Looking at the attacker's second option, random noise will always violate the changes caused by the prior attack, to a greater or lesser degree. In order for the random noise to have less effect on the changes caused by the attack, the attack itself would have to cause greater changes in the image, but this is where the need to check the random noise at different intensities comes from. However, the use of an adversarial attack with a significantly higher epsilon is also limited by the fact that the noise can be spotted by the human eye or any kind of noise detector,

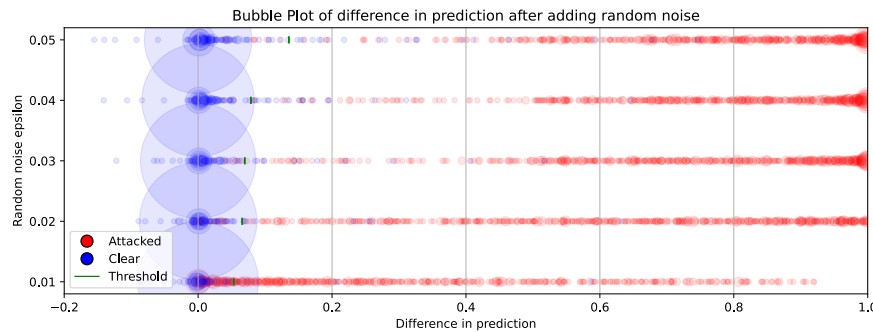

Figure 4: Bubble Plot of difference in prediction after adding random noise for InceptionV3-based model trained on Brain Tumor dataset for BIM attack.

Table 2: Detection results [%] for attacked and clean images across datasets and models for the best epsilon of random noise and standard threshold $\theta = 2 \cdot \mathrm{std}(t)$.

| Attack | ResNet50V2 | | | InceptionV3 | | | ViT-B16 | | | |
| | Chest X-Ray | Brain MRI | Fundus | Chest X-Ray | Brain MRI | Fundus | Chest X-Ray | Brain MRI | Fundus | MEAN |
|---|---|---|---|---|---|---|---|---|---|---|
| FGSM | 98.8 \| 3.2 | 95.2 \| 3.0 | 81.7 \| 4.6 | 95.7 \| 5.2 | 97.9 \| 2.6 | 94.2 \| 3.6 | 95.0 \| 4.3 | 95.7 \| 4.7 | 93.5 \| 4.3 | 94.2 \| 3.9 |
| BIM | 99.3 \| 2.9 | 98.3 \| 2.0 | 96.3 \| 4.7 | 95.8 \| 4.0 | 98.7 \| 1.9 | 93.7 \| 3.6 | 96.4 \| 6.3 | 97.0 \| 4.4 | 96.0 \| 3.0 | 96.8 \| 3.6 |
| PGD | 99.3 \| 2.9 | 98.3 \| 2.8 | 97.7 \| 4.4 | 96.7 \| 3.5 | 98.8 \| 1.7 | 94.4 \| 4.5 | 95.4 \| 5.9 | 98.4 \| 5.3 | 96.0 \| 4.2 | 97.2 \| 3.9 |
| DF | 98.8 \| 1.0 | 99.8 \| 1.1 | 96.2 \| 3.5 | 97.0 \| 2.0 | 96.8 \| 0.9 | 92.6 \| 5.9 | 93.0 \| 3.2 | 98.3 \| 1.9 | 90.9 \| 4.3 | 95.9 \| 2.6 |
| MEAN | 99.1 \| 2.5 | 97.9 \| 2.2 | 93.0 \| 4.3 | 96.3 \| 3.7 | 98.1 \| 1.8 | 93.7 \| 4.4 | 95.0 \| 4.9 | 97.4 \| 4.1 | 94.1 \| 3.9 | 96.1 \| 3.5 |

and also resulting in a very extreme prediction that will raise suspicion. All of the aforementioned arguments make selecting an effective attack method an extremely difficult task, making the defense method still effective. In addition, the proposed method is much simpler to implement, as it does not require the creation of an additional model. It is worth noting the computational cost, which is less computationally expensive to the iterative attack because the attack requires checking the prediction if the class has changed, pulling the gradient to select the perturbation, and the operation of adding it to the image. In comparison, the defense involves only adding random noise and checking the prediction of the image being checked, and checking the prediction after adding the noise. So, the computational cost is mainly dependent on the number of attempts of adding noise, which is to make sure that the randomly selected noise is not an extreme case that can have a huge impact on the prediction of the unattacked image.

### 3.3 STATISTICAL ANALYSIS OF DETECTION ACCURACY

In Fig. 4 we can observe a clear distinction in the prediction differences between attacked and clean images. The distribution of clean cases closely follows a Sigmoid function, which confirms that random noise changes the prediction of a clean image without any direction. On the other hand, in the attacked cases, practically no image returned the direction of prediction to its class, and it was significantly changed towards the original class of the image. However, for a more complete picture of the effectiveness of this method, it was useful to exclude the images originally mispredicted by the model. The number of attacks that had a relatively small difference between the prediction of the attacked and the noisy attack image decreased significantly, making the differences more visible.

Considering that there can be no mistakes in medicine, it can be assumed that the prediction of the model is taken into account only in cases where the model has high confidence in the predicted class. It should be noted that images at the threshold of model certainty are more susceptible to potential attacks and also are more sensitive to the imposition of random noise. At the same time, to succeed in an adversarial attack, the difference in prediction with class change could be significantly smaller, so the effectiveness of the method was also tested for cases where the prediction was more certain.

Valuable results clarified on average for the prediction difference threshold using the standard deviation. As we can see in Table 2, the results of the algorithm look very promising. On average, for the selected threshold, it manages to separate practically all the attacked images, with a very

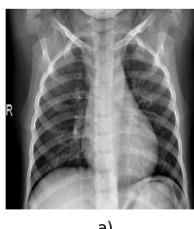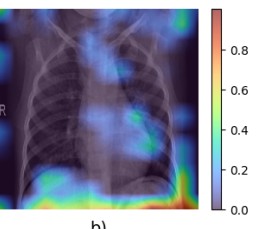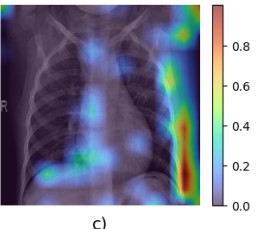

a)    b)    c)

Figure 5: Visualization of adversarial impact on model interpretation: a) Original clean Chest X-ray image, b) Heatmap of the clean image, highlighting the model's focus, c) Heatmap of the adversarially attacked image, showing shifts in attention.

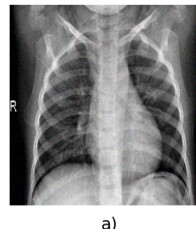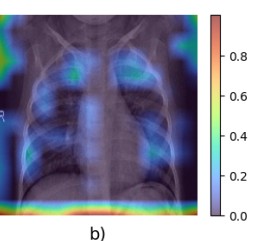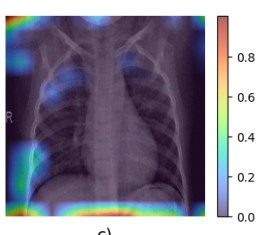

a)    b)    c)

Figure 6: Effect of adversarial attacks on a noise-perturbed image: a) Chest X-ray image with randomly added noise, b) Heatmap of the noisy image, showing the model's attention, c) Heatmap of the adversarially attacked noisy image, illustrating how the attack influenced the model's focus.

low percentage of clean images. By manually selecting the threshold for a particular model, the effectiveness of detection can be further increased. Together with the discarding of images with low confidence, it can detect as much as 100% of the attacked images, making the algorithm very valuable for medical systems as well as for all others based on deep neural networks.

As shown in Fig. 3, the impact of noise and adversarial attacks on Sigmoid performance is presented, demonstrating how both noise and adversarial perturbations influence the model's confidence in its predictions. Furthermore, we also analyzed the adversarial impact on model interpretation. Fig. 5 shows how adversarial attacks alter the interpretation of the model. The heatmap comparison reveals a shift in focus after the attack, demonstrating its influence on decision-making. Fig. 6 extends this analysis by introducing random noise before the attack. Although the model still maintains focus on key regions with noise alone, the adversarial attack significantly disrupts its attention.

## 4 CONCLUSION

This study examines how adversarial attacks affect medical image classification, showing that perturbations, whether random noise or targeted attacks, can influence both model accuracy and interpretability. By analyzing prediction differences across clean, noisy, and adversarially attacked images, we demonstrate the risks these attacks pose to deep learning models in medical imaging, especially when the attacker has direct access to the model architecture. Our solution shows how to defend against white-box adversarial attacks with unusually high detection rates. These solutions represent the future of medical systems, allowing them to rely on deep neural networks without fear of data falsification by adversarial attacks.

### 4.1 FUTURE RESEARCH DIRECTIONS

Future work will focus on strengthening defense mechanisms to improve model robustness. Expanding the study to larger datasets and different imaging modalities will help assess the broader applicability of these findings. Especially for multi-class datasets to make it plausible that adding random noise will result in a return to the previous class. In addition, improving interpretability techniques will be essential to ensure reliable AI-assisted medical diagnoses.

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
