# OpenReview forum: "Robust Detection of Directional Adversarial Attacks in Deep Neural Networks for Radiological Imaging"
_ICLR.cc/2026/Conference — Submitted to ICLR 2026_

### Official Review · Reviewer_AWUZ · 2025-10-28

**Soundness:** 2
**Presentation:** 2
**Contribution:** 2
**Rating:** 2
**Confidence:** 4

**Summary:**

In the field of radiology, deep neural network (DNN) models have been widely used to detect abnormalities in radiographic images. However, these models can be easily misled by carefully crafted adversarially manipulated samples, revealing a lack of robustness. To address this issue, this paper proposes a simple yet effective method for detecting adversarial-attacked samples. The method performs a “re-attack” on an unknown sample by adding noise of a specific intensity and analyzes the change in the model’s final prediction probabilities to distinguish clean samples from attacked ones. Experiments conducted on three mainstream radiographic image datasets, using various types of adversarial samples (FGSM, BIM, etc.), demonstrate the effectiveness of the proposed method. The results show that their approach can successfully differentiate adversarial-attacked samples from clean samples, exhibiting impressive performance and strong potential for clinical application.

**Strengths:**

1. The proposed method is simple, effective, and training-free. Therefore, it can be seamlessly integrated with existing neural network–based radiographic anomaly detection approaches, offering strong generalizability and practical potential.
2. The proposed method demonstrates robust performance against various types of adversarial attacks, highlighting its strong resilience.
3. The authors providing an interpretation of their proposed method’s effectiveness from the perspective of representation learning, for heatmap visualizations are used to illustrate how the method influences deep learning models.

**Weaknesses:**

Although this paper proposes a seemingly simple and effective method for detecting adversarial attacks on radiographic images, I believe it still has several weaknesses. The main issues are as follows:
1. The authors’ research motivation and the scientific questions to be addressed should be clearly defined and stated in the Introduction of the paper. E.g: “...(Based on the above practical considerations), distinguishing between clean samples and adversarial samples is a research problem worth exploring in (related research)”.
2. Although Section 1.2 is devoted to discussing adversarial attacks in medical imaging, there are relatively few references that simultaneously address both key themes: medical imaging and adversarial attacks. This section appears to focus more on adversarial-attack-related content rather than on their applications in medical imaging.
3. ICLR focuses on representation learning, but the representation-learning approach proposed in this paper appears to be limited to feeding medical images into a pre-trained network — a representation-learning strategy that seems too simplistic for ICLR’s scope. Besides, although ICLR also welcomes simple and effective methods, this paper lacks broader and deeper experiments and analyses of its proposed method.
4. The proposed method appears to require setting noise intensity thresholds and prediction-difference thresholds specific to a particular dataset, neural network, and attack type, which may hinder its generalization to different Out-of-Distribution medical imaging data.
5. This paper contains many ambiguous descriptions, and some figure and table captions lack sufficient explanation. The presentation of data is also not standardized (e.g., using long blocks of text instead of tables or charts), which seriously affects the readability of the paper. For details, please refer to the questions I raised in the Questions section. Note that the issues I raised represent only a portion of the problems in this paper. The authors should carefully examine this particular weakness.

**Questions:**

1. The case study shows in Figure 1 requires further improvement. It should provide a more detailed explanation of the potential outcomes that such adversarial samples may cause within the network.
2. All external data should be accompanied by reliable source citations. E.g. in Section 1.1: “In 2022 alone, more than 28 million patient records were exposed in healthcare-related data breaches in the US, with imaging data often among compromised content.” Moreover, the related research methods discussed in Section 1.4 are outdated and should be supplemented with more recent studies.
3. It appears that the Related Works section could be divided into two parts: “Applications of adversarial attacks in medical imaging” and “Methods for defending against attacks in medical imaging.” To improve readability, these two parts should be more clearly separated. I suggest that Section 1.2 could focus on adversarial attacks, while Section 1.3 could focus on defense methods. Moreover, incorporating the medical imaging–related content into the Related Works section might be more appropriate.
4. This paper lacks broader and deeper experiments and analyses of the approach. E.g. (1) Is this strategy effective for other visual tasks in medical imaging, such as detection and segmentation? (2) The datasets used in this paper appear to be binary-classification problems, does the strategy remain effective for multi-class problems? (3) A deeper, more persuasive analysis is needed to explain why the proposed method works.
5. In Section 2.4, although the authors propose manually setting a threshold for prediction differences, the meaning of the parameter $\theta$ remains unclear, making Table 2 extremely difficult to interpret. Is the author implying that samples exceeding $\theta$ are considered anomalous, or that samples exceeding the manually set prediction-difference threshold are? In Table 2, does the second subcolumn of each column represent $\theta$ or the manually defined prediction-difference threshold?
6. In Section 3.2, DF causes the prediction accuracies of ResNet-50, ViT-B16, and Inception-V3 to change by 99%, 99%, and 62.6%, respectively. What types of networks were used for the other attack methods, and what specific changes occurred? It would be clearer to present this information in a table.
7. In the 3rd paragraph of Section 3.2, the authors mentioned that “The most effective universal power for all types of models was 3%.” However, they subsequently mentioned that “So, the added changes to the image on average were smaller than for the other types of attack, resulting in a best epsilon of random noise for each model attacked by DeepFool was 0.01.” Is there a contradiction here, or are the referents of the two unclear? In addition, the experimental results in this section should be presented in a table to improve readability.
8. The caption of the bubble plot presents in Figure 4 is unclear. For example, what do the radii of the bubbles represent? Does each bubble correspond to a single sample?
9. It seems that the Brain Tumor dataset is merely a binary classification problem (healthy vs. tumor). Therefore, the conclusion mentioned in Section 3.3 is rather obvious: the predictions of most adversarial samples tend to move toward the correct class, since there are only two possible labels in this context. What I am more concerned about is whether the proposed method can ensure that, in multi-class tasks, the predictions of attacked samples still move toward the correct class—either uniquely or toward the majority of correct classes among many categories.
10. Although this paper focuses on defending against adversarial attacks and proposes a simple and effective method for detecting adversarial examples, similar studies should exist in the broader image domain. In Section 3.3 & Table 2, the paper does not seem to compare its approach with other image-based adversarial detection methods, making the baseline unclear.

---

### Official Review · Reviewer_i1wG · 2025-10-30

**Soundness:** 2
**Presentation:** 2
**Contribution:** 2
**Rating:** 2
**Confidence:** 4

**Summary:**

The paper focuses on adversarial example detection that leverages subsequent attacks using random noise. Specifically, the paper focuses on the Chest X-ray for computer-aided medical image analysis. The detection primarily relies on the prediction gap between clean/adversarial images and their noise-perturbed counterparts. In addition to the chest X-ray image context, the paper further explores the generalization of the proposed method in the context of eye fundoscopy and brain MRI. Experiments are also conducted across diverse vision backbones.

**Strengths:**

1. The topic is interesting. Investigating adversarial threats and their countermeasures in the context of medical image analysis is important in the current society.
2. The visualizations are well presented and organized, which visually support the claim that small noise nudges attacked images back toward the clean class focus.
3. The paper explores diverse diagnosis tasks in the context of medical image analysis. Furthermore, experimental results across diverse backbones are given.

**Weaknesses:**

1. The motivation seems to be weak. Although it might be intuitive that clean samples and their adversarial counterparts can exhibit different behavior when encountering noise. However, an empirical analysis on a certain dataset or some theoretical analyses would further improve the paper. Otherwise, the correctness of the motivation is not justified.
2. The visualizations of Figure can further be improved. The authors can consider enhancing the pixel intensity for adversarial noise to get a better visualization. In addition, some illustrative signs can also enhance the motivation.
3. AutoAttack and adaptive attack results are missing. In addition, robustness against higher attack strength (perturbation radii) should be given.
4. It appears that no equations are given in this paper. The paper would benefit more from mathematical analyses.
5. The paper lacks explicit and empirical comparisons with adversarial fine-tuning and adversarial purification.

**Questions:**

1. Can the proposed method be extended to multimodal architectures? In other words, would the proposed detection method still be applicable to multimodal large language models, e.g., CLIP or medical CLIP?
2. Would the proposed method be improved via an adaptive (learned) detection threshold compared with a fixed threshold?
3. I would suggest adding an additional discussion about the practicality of adversarial attacks for medical image analysis. Because medical diagnosis machines are typically offline. Then how would the attacker create adversarial medical examples?

---

### Official Review · Reviewer_Jp2R · 2025-10-30

**Soundness:** 1
**Presentation:** 2
**Contribution:** 2
**Rating:** 2
**Confidence:** 4

**Summary:**

This paper introduces a framework for detecting directional adversarial attacks in medical imaging models. The approach compares model predictions before and after adding small random noise, assuming clean images are more stable under such perturbations than adversarial ones. Experiments on three medical datasets (ChestX-Ray, Brain MRI, Retinal Fundus) using ResNet50V2, InceptionV3, and ViT-B16 report detection accuracies up to 99.8% with low false positives.

**Strengths:**

1.	The paper addresses an important and safety-critical problem: detecting adversarial manipulations in medical imaging models used for diagnosis. With deep learning increasingly adopted in clinical practice, ensuring reliability against such attacks is both timely and significant.
2.	The proposed method uses only test-time perturbations and prediction differences, making it easy to add as a post-hoc verification step without retraining or model changes. This design fits well with real world medical AI deployment and regulatory constraints.

**Weaknesses:**

1.	Limited Novelty: The method is very close to earlier prediction-difference and stability-based detection approaches [1, 2], which already use small perturbations to test model confidence. The claimed focus on “directional attacks” adds little new, since the method does not actually model or exploit directionality-adding random noise and thresholding are general operations.

2.	Insufficient Theoretical Foundation: The idea that random noise can “neutralize” adversarial perturbations is not theoretically justified. No analytical proof or clear reasoning shows that unstructured noise can reliably counter structured attacks, making this assumption largely heuristic [3].

3.	Experimental Design Concerns: Reported accuracies above 99 % seem implausible for this task and are not well-supported by methodological details. It is unclear how thresholds were tuned, raising possible data leakage concerns. Standard robustness metrics (e.g., AUROC, TPR@5%FPR) and variance reporting are missing, which makes the results hard to trust.

4.	Lack of Analytical Depth: The paper provides almost no formal analysis of why prediction differences should distinguish clean from adversarial samples. The math is descriptive rather than explanatory, unlike prior work such as Mahalanobis or energy-based detectors that include probabilistic reasoning.

[1] Guo, Feng, et al. "Detecting adversarial examples via prediction difference for deep neural networks." Information Sciences 501 (2019): 182-192.

[2] Lee, Kimin, et al. "A simple unified framework for detecting out-of-distribution samples and adversarial attacks." Advances in neural information processing systems 31 (2018).

[3] Goodfellow, Ian J., Jonathon Shlens, and Christian Szegedy. "Explaining and harnessing adversarial examples." arXiv preprint arXiv:1412.6572 (2014).

**Questions:**

1.	Could author provide how sensitive is the method to the choice of random noise strength (ε)? Would slightly different noise magnitudes or distributions (e.g., Gaussian vs. uniform) significantly change performance?
2.	How would your method perform against stronger recent detection baselines (e.g., Energy-based OOD detection, or LogitNorm)?

---

### Official Review · Reviewer_hi9m · 2025-10-30

**Soundness:** 3
**Presentation:** 3
**Contribution:** 3
**Rating:** 4
**Confidence:** 4

**Summary:**

This paper proposes a lightweight adversarial detection framework for medical imaging systems that identifies directional adversarial attacks that intentionally mislead models toward a specific incorrect class. The proposed method introduces a random noise–based detection mechanism: it analyzes prediction differences between clean vs. noisy-clean images, and adversarially attacked vs. noisy-attacked images, to distinguish genuine from manipulated images. Experiments were conducted on three datasets, using multiple deep learning backbones under common adversarial attacks.

**Strengths:**

The framework’s core idea, leveraging prediction variance after random perturbation, is elegant and training-free. It cleverly repurposes noise sensitivity as a proxy for adversarial trace detection without retraining models.

Low false-positive rates (≈ 2–4%) make it practical for medical deployment.

Covers four adversarial attacks (FGSM, BIM, PGD, DeepFool) under multiple epsilon values and models.

Incorporates datasets from major imaging modalities (X-ray, MRI, fundus).

**Weaknesses:**

The approach is largely empirical; no theoretical guarantee or formal proof explains why random noise restores original class predictions under adversarial perturbations.

Tested only on 2D image classification; not validated for large 3D imaging datasets (CT volumes, fMRI). Multi-class or regression tasks (e.g., lesion segmentation) are untested.

The fixed threshold may be suboptimal for other datasets or attack strengths, which could impact generalization.

**Questions:**

na

---

### Meta-Review · Area_Chair_B3ga · 2026-01-08

**Summary:**

The core concerns that informed this decision are:

Novelty and Theoretical Foundation (Critical):
Multiple reviewers (Jp2R, i1wG, AWUZ) identified limited novelty, noting strong similarity to existing prediction-difference methods [Guo et al., 2019; Lee et al., 2018]
Lack of theoretical justification for why random noise can neutralize adversarial perturbations (Jp2R, i1wG)
The "directional attack" focus adds minimal conceptual advancement beyond general perturbation-based detection

Experimental Rigor (Critical):
Implausibly high detection accuracies (>99%) without sufficient methodological transparency (Jp2R)
Missing variance reporting, AUROC, and standard robustness metrics (Jp2R)
Potential data leakage concerns regarding threshold tuning (Jp2R)
Absence of adaptive attacks and AutoAttack evaluation (i1wG)
No comparison with recent strong baselines like Energy-based OOD or LogitNorm (Jp2R)

**Reviewer Concerns:**

The author did not provide a rebuttal.

**Reviewer Scores:**

The author did not provide a rebuttal.

---

### Decision · Program_Chairs · 2026-01-26

Reject